# Significance of the drug-induced lymphocyte stimulation test for various oral mesalamines in ulcerative colitis with mesalamine intolerance

**Akira Madarame** [ID][☉][*], **Masakatsu Fukuzawa**[☉], **Kumiko Uchida, Tadashi Ichimiya, Sakiko Naito, Yoshiya Yamauchi, Takashi Morise, Yasuyuki Kagawa, Takahiro Muramatsu, Takao Itoi**

Department of Gastroenterology and Hepatology, Tokyo Medical University Hospital, Tokyo, Japan

☉ These authors contributed equally to this work.
* madarame@tokyo-med.ac.jp

## Abstract

This study aimed to determine if the drug-induced lymphocyte stimulation test (DLST) for various oral mesalamines can diagnose mesalamine intolerance and predict the success of retreatment in patients with adverse events (AEs) due to the first oral 5-aminosalicylate (5-ASA) formulations administered. Data from patients with ulcerative colitis who experienced AEs after administration of the first oral 5-ASA and underwent DLSTs for two or more types of mesalamine, including the first oral 5-ASA were retrospectively analyzed. Mesalamine intolerance was defined as AEs within 6 months of starting the first oral 5-ASA and the inability to take oral mesalamine. Clinical characteristics, symptoms, type of first oral 5-ASA, DLST results, and the efficacy of retreatment with oral mesalamine were compared. The DLST for the first oral 5-ASA (F-DLST), highest DLST among the different oral mesalamine types (H-DLST), and mean DLST (M-DLST) were analyzed. Twenty-eight patients (median age 39 years, 57.1% male) were eligible; six patients were tolerant to oral mesalamine and 22 were intolerant. Positive F-DLST (odds ratio [OR], 2.300; p = 0.002), positive M-DLST (OR, 2.667; p = 0.007), and older age at diagnosis (median 24.5 vs. 41.5; p = 0.006) were associated with mesalamine intolerance. Fourteen of the 28 patients underwent mesalamine retreatment. Higher F-DLST (median 88.0 vs. 174.0; p = 0.026), M-DLST (median 118.5 vs. 170.3; p = 0.040), and older age at diagnosis (median 24.5 vs. 39.0; p = 0.033) were associated with retreatment failure for oral mesalamine. DLSTs for various oral mesalamine formulations may be useful in predicting mesalamine intolerance and retreatment outcomes. However, their clinical utility should be interpreted with caution due to the risk of false-positive and false-negative results.

**Data availability statement:** The minimal dataset required to replicate the study's findings is fully contained within the paper.

**Funding:** The author(s) received no specific funding for this work.

**Competing interests:** The authors have declared that no competing interests exist.

## Introduction

Formulations of 5-aminosalicylate (5-ASA) are the first-line treatment for patients with mild to moderate ulcerative colitis (UC); 5-ASAs, such as mesalamine, induce and maintain UC remission and prevent colorectal cancer [1,2]. Salazosulfapyridine (SASP) is a sulfonamide-based prodrug that can be used to treat UC. It delivers 5-ASA to the colon through bacterial azo-reduction; however, it contains a sulfapyridine moiety, so SASP differs chemically and pharmacokinetically from pure mesalazine formulations and may cause additional adverse effects, including headache, skin rash, and male infertility. Therefore, mesalamine, a 5-ASA without the sulfapyridine group, was developed. Mesalamine is tolerated better than SASP [3,4]. The efficacy of 5-ASA depends on the concentration in the colonic mucosa. Therefore, 5-ASA formulations require drug delivery systems to deliver the drug to the colon efficiently. In Japan, the approved mesalamine-based 5-ASA formulations include Pentasa® with a time-dependent release system, Asacol® with a pH-dependent colon-targeted oral drug delivery system, and Lialda® with a multi-matrix system.

Mesalamine-based 5-ASA formulations are well tolerated but can cause symptoms similar to UC exacerbations, including fever, abdominal pain, diarrhea, and hematochezia [5–7]. Mesalamine intolerance may also cause organ damage such as pericarditis and pneumonia [8–11]. Hiraoka et al. reported that the incidence of adverse events (AEs) related to 5-ASAs more than tripled from 5.3% in 2007–2010–16.2% in 2014–2016 [12]. According to a study by Hibiya et al., a higher percentage of patients with mesalamine intolerance underwent colectomies compared with the percentage of patients tolerating mesalamine who underwent colectomies (Hazard ratio: 4.92; 95% confidence interval: 2.58–9.38) [13]. Patients who experience 5-ASA-induced AEs can be retreated after desensitization therapy or treated with another mesalamine-based ASA [14–16]; however, the AEs may recur [17]. Although mesalamine intolerance sometimes causes serious AEs, methods for diagnosing mesalamine intolerance have not been established.

The drug-induced lymphocyte stimulation test (DLST) measures the uptake of $^3$H-thymidine by proliferating lymphocytes after stimulation with the target drug [18]. The accuracy of DLST in diagnosing drug allergies varies widely depending on the target drug [19,20], and the low sensitivity of DLST for detecting AEs related to 5-ASA is problematic [21,22]. The accuracy of the DLST for mesalamine affects decisions concerning retreatment with mesalamine. False positives may develop due to an allergy to the excipients of the mesalamine [23], and the opportunity for retreatment with mesalamine may be lost. Several studies have highlighted the high success rate of retreatment with mesalamine, even with a positive DLST for the suspect drug [24,25]. However, Shimizu et al. reported that five patients with positive DLSTs for 5-ASA were rechallenged with 5-ASA, and all patients failed [26]. In previous studies, DLST was performed only for the suspected 5-ASA agent, and the association between DLST results for various oral mesalamine formulations and mesalamine intolerance remains unclear. We hypothesized that testing various mesalamine formulations using DLST would improve the diagnostic sensitivity of mesalamine

intolerance. Therefore, this study aimed to evaluate the relationship between DLST results for various oral mesalamine, mesalamine intolerance, and retreatment outcomes.

## Materials and methods

### Patients

Consecutive UC patients treated from 2014 to 2021 at the Department of Gastroenterology and Hepatology, Tokyo Medical University (Tokyo, Japan) were enrolled in the study. All patients were at least 18 years old and were diagnosed with UC based on standard clinical, endoscopic, and historical criteria according to the Japanese Research Committee on Inflammatory Bowel Disease [27]. Patients were prescribed Pentasa® (Kyorin Pharmaceutical Co., Ltd., Tokyo, Japan), Asacol® (Zeria Pharmaceutical Co., Ltd., Tokyo, Japan), Lialda ® (Mochida Pharmaceutical Co., Ltd., Tokyo, Japan), or Salazopyrin® (Pfizer Co., Ltd., Tokyo, Japan). Patients who experienced AEs after the first oral 5-ASA and underwent DLSTs for two or more types of mesalamine, including the first oral 5-ASA, were identified. Patients with unknown disease extent or type of oral mesalamine at diagnosis, patients who started oral SASP at diagnosis, patients receiving topical mesalamine before oral mesalamine, and patients with a history of hospital visits of less than 6 months were excluded from the study. Patients with a history of using topical 5-ASA formulations were excluded because this study specifically aimed to evaluate mesalamine intolerance and retreatment outcomes associated with oral formulations. The inclusion of patients who experienced AEs only with topical 5-ASA preparations may introduce heterogeneity because their pharmacokinetics and routes of exposure are different from those of oral formulations. Therefore, patients who had prior topical 5-ASA use were excluded to maintain the consistency and validity of the study cohort. A retrospective chart review was performed by two gastroenterologists (AM and MF). Any discrepancies were resolved by discussion with the senior author.

### Data collection and definitions

Electronic medical records and the UC database were reviewed to ensure the inclusion of all patients. The clinical characteristics of patients with AEs to the first oral 5-ASA were extracted from medical records, including age at diagnosis, extent of disease at diagnosis of UC, type of first oral 5-ASA, clinical symptoms of AEs due to first oral 5-ASA, and duration of AEs due to first oral 5-ASA.

AEs due to the first oral 5-ASA were defined as symptoms occurring after receiving the first oral 5-ASA that resolved after discontinuing the drug, with or without corticosteroid treatment. Mesalamine intolerance was defined as (1) the occurrence of AEs within 6 months of starting the first oral 5-ASA and the inability to continue taking oral mesalamines owing to intolerance upon retreatment or (2) patients whose initial AEs were severe or involved significant organ toxicity and therefore did not receive mesalamine retreatment. Mesalamine tolerance was defined as patients tolerating at least one oral mesalamine [13]. Patients who experienced only mild AEs from the first oral 5-ASA and had no organ damage received mesalamine retreatment after consultation with their attending physician. Retreatment success was defined as the ability to continue taking at least one oral mesalamine formulation for >6 months after starting the first oral 5-ASA without AEs, whereas retreatment failure was defined as discontinuation of retreatment with mesalamine owing to AEs. Patients who achieved successful retreatment were ultimately assigned to the mesalamine-tolerant group, while those in whom retreatment failed were assigned to the mesalamine-intolerant group.

DLST was performed by SRL Inc. (Hachioji, Japan). For each DLST performed per drug, 5 mL of whole blood was collected and transferred to a heparinized blood collection tube. Lymphocytes were isolated from the whole blood through density gradient centrifugation and suspended in the RPMI1640 medium (DS Pharma Biomedical, Osaka, Japan). The cells were cultured with mesalamine for 72 h. ³H-Thymidine was then added, and incubation was continued for an additional 16 h. The uptake of $^3$H-thymidine, which was used as a control, in lymphocytes not exposed to mesalamine was measured. The stimulation index (SI) was defined as the ratio of $^3$H-thymidine uptake between the mesalamine-treated and control samples [18]. Positive DLST was defined as SI > 180%. The SI of the first oral 5-ASA was defined as F-DLST,

the highest SI among the various types of oral mesalamines was defined as H-DLST, and the M-DLST was defined as the mean SI for the different mesalamines.

## Study design and statistical analyses

The study was a single-center retrospective cohort study. Patients with UC who experienced AEs to the first oral 5-ASA were divided into two groups: (a) mesalamine-tolerant group: patients who were tolerant to another oral mesalamine and (b) mesalamine-intolerant group: patients who were intolerant to one or more oral mesalamines and discontinued mesalamine treatment. Patient characteristics and the medical course were compared using the χ2, Fisher's exact, or Mann–Whitney U-tests. DLST was selected as a quantitative variable, and missing DLST data were removed. Univariate analyses were performed to identify factors associated with tolerance to retreatment with mesalamine. The cutoff values for DLST in predicting mesalamine intolerance and the efficacy of retreatment with oral mesalamine were examined using receiver operating characteristic (ROC) analysis. Odds ratios (ORs) with 95% confidence intervals (CIs) were calculated. P values <0.05 were considered statistically significant. All statistical analyses were performed using IBM SPSS Statistics version 29.0.1.0 (IBM Corp., Armonk, N.Y., USA).

## Ethical considerations

The Ethics Committee of Tokyo Medical University School of Medicine approved this study (approval number: T2021-0352) on February 25, 2022, and the study was conducted in accordance with the tenets of the Declaration of Helsinki. Medical records and UC database were accessed for research purposes between February 25, 2022 and September 1, 2022. The authors had access to identifiable participant information during data collection; however, all data were anonymized before analysis. Moreover, this study was conducted using an opt-out approach approved by the institutional ethics committee, and the requirement for written informed consent was waived in accordance with relevant ethical guidelines. Furthermore, information about this study, including the opt-out process, was publicly posted at Tokyo Medical University Hospital, and no patients objected to the anonymized use of their data.

## Results

### Patient characteristics

Between 2014 and 2021, 695 patients with UC visited our hospital. Seventy-four patients were excluded due to the lack of detailed information, and 13 were excluded owing to prior use of topical mesalamine preparations before initiating oral 5-ASA. Of the remaining patients, 564 (92.8%) had no AEs due to the first oral 5-ASA and 44 (7.2%) had AEs. Moreover, 28 of the 44 patients underwent DLSTs for two or more types of mesalamine, including the first oral 5-ASA (Fig 1). None of the patients had a history of hospital visits of <6 months. Of the 28 patients who underwent DLST, 6 were classified into the mesalamine-tolerant group and 22 into the mesalamine-intolerant group. Table 1 shows the clinical characteristics of the patients with AEs due to the first oral 5-ASA. The median age at diagnosis of UC was 39 years (interquartile range [IQR]: 27.5–50.5), and 16 (57.1%) of the patients were male. At the time of UC diagnosis, 18 patients had extensive colitis, five patients had left-sided colitis, and five patients had proctitis. The median first oral 5-ASA dose was 4,000 mg (IQR: 3,600–4,800). The first oral 5-ASAs were time-dependent in eight cases, pH-dependent in six cases, and multimatrix in 14 cases. The time to the onset of AEs due to the first oral 5-ASA was 13 days (IQR: 9–22.5). The most common symptoms were fever (n = 17, 60.7%), diarrhea/bloody stools (n = 13, 46.4%), and abdominal pain (n = 7, 25.0%). Three patients were affected: one had pneumonia, another experienced myelosuppression, and the third suffered from both pancreatitis and liver dysfunction.

### Comparison of the mesalamine-intolerant and tolerant groups

Table 2 shows the comparison of clinical backgrounds between the mesalamine-intolerant and tolerant groups. The mesalamine-intolerant group was younger than the tolerant group by age at diagnosis of UC (23.5 vs 41.5 years, p = 0.006).

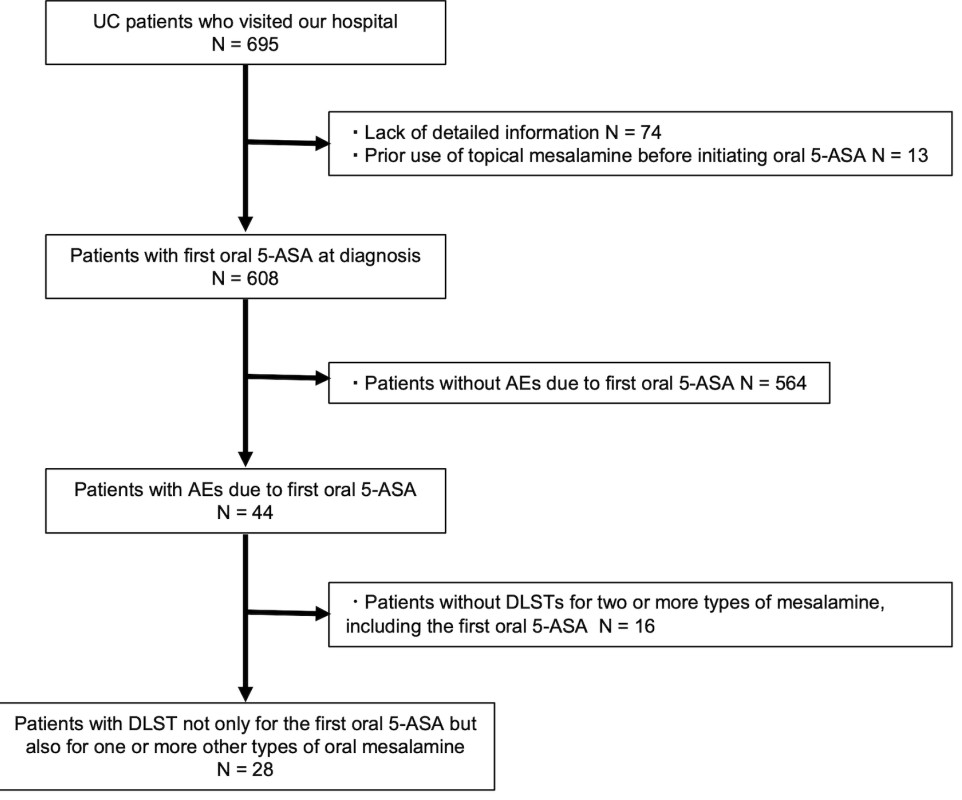

**Fig 1. Flowchart of eligible patients.**

No significant differences in sex, extent of disease at diagnosis of UC, or type of first oral 5-ASA were detected between the two groups. In addition, no significant differences in the AE symptoms due to the first oral 5-ASA (diarrheal bloody stools, fever up, abdominal pain, nausea, headache, fatigue, skin rash, joint pain, pancreatitis, pneumonia, bone marrow suppression, liver dysfunction) or the time to onset of AEs due to the first oral 5-ASA were detected between the groups.

## Relationship between DLST and mesalamine intolerance

In the 28 patients who underwent F-DLSTs, 15 (53.6%) tests were negative and 13 (46.4%) tests were positive. All 13 patients with positive F-DLSTs were mesalamine-intolerant group (p = 0.018, OR 2.444, 95% CI 1.479–4.039) (Table 3). The median F-DLST was higher in the mesalamine-intolerant group than in the mesalamine-tolerant group (250.5% vs. 83.5%, p = 0.002).

DLSTs for various oral mesalamines, including the first oral 5-ASA, were performed in 28 patients. Four patients were tested for one type of oral mesalamine, four patients were tested for two types, 11 patients were tested for three types, and 13 patients were tested for four types. Eight cases (28.6%) were negative and 20 cases (71.4%) were positive for the H-DLST. Eighteen of the 20 patients with positive H-DLSTs were intolerant to oral mesalamines, and the association was significant (p = 0.038, OR 9.000, 95% CI 1.201–67.417). The median H-DLST was higher in the mesalamine-intolerant group than the median H-DLST in the mesalamine-tolerant group (411.0% vs. 153.5%, p = 0.005).

M-DLSTs were positive in 15 patients and negative in 13 patients. The 15 patients with positive M-DLSTs were all intolerant to oral mesalamines (p = 0.005, OR 3.143, 95% CI 1.705–5.794). The M-DLSTs were higher in the mesalamine-intolerant group than the M-DLSTs in the mesalamine-tolerant group (311.1% vs. 110.4%, p = 0.001).

**Table 1. Clinical characteristics of patients with adverse events due to the first oral 5-ASA.**

| Total number | 28 |
|---|---|
| Sex (female/male) | 12/16 |
| Age at diagnosis of UC (years); IQR | 39 (27.5–50.5) |
| Extent of UC at the time of diagnosis (%) | |
| Extensive | 18 (64.3) |
| Left-sided | 5 (17.9) |
| Rectum | 5 (17.9) |
| First oral mesalamine (%) | |
| Time-dependent release | 8 (28.6) |
| pH-dependent release | 6 (21.4) |
| Multimatrix | 14 (50.0) |
| Median first oral 5-ASA dose (mg); IQR | 4,000 (3,600–4,800) |
| Symptoms of adverse events due to the first oral 5-ASA (%) | |
| Fever | 17 (60.7) |
| Diarrhea and bloody stools | 13 (46.4) |
| Abdominal pain | 7 (25.0) |
| Joint pain | 4 (14.3) |
| Nausea | 2 (7.1) |
| Headache | 1 (3.6) |
| Fatigue | 1 (3.6) |
| Skin rash | 1 (3.6) |
| Pancreatitis | 1 (3.6) |
| Pneumonia | 1 (3.6) |
| Bone marrow suppression | 1 (3.6) |
| Liver dysfunction | 1 (3.6) |
| Time to the onset of adverse events due to the first oral 5-ASA (days); IQR | 13 (9–22.5) |

Values are presented as medians or numbers. 5-ASA, 5-aminosalicylate; IQR, interquartile range; UC, ulcerative colitis.

The sensitivity and specificity of a positive F-DLST for predicting mesalamine intolerance were 0.591 and 1.00; the positive and negative predictive values were 1.00 and 0.400. Positive H-DLSTs had a sensitivity of 0.818, a specificity of 0.667, a positive predictive value of 0.900, and a negative predictive value of 0.500. Positive M-DLSTs had a sensitivity of 0.682, a specificity of 1.00, a positive predictive value of 1.00, and a negative predictive value of 0.462.

### Retreatment with oral mesalamine after AEs due to the first oral 5-ASA

Fourteen patients underwent retreatment with oral mesalamines after AEs to the first 5-ASA (Fig 2), of whom six achieved retreatment success and were classified into the mesalamine-tolerant group, whereas eight experienced retreatment failure and were classified into the mesalamine-intolerant group. Second, third, and fourth-line therapies with oral mesalamine achieved success rates of 21.4% (three of 14), 60.0% (three of five), and 0% (none of one), respectively. For the second-line therapy, eight patients were switched to a different type of oral 5-ASA preparation. Four patients underwent desensitization therapy with a time-dependent release type of 5-ASA that differed from their initial formulation. One patient underwent desensitization therapy using the same type of time-dependent release 5-ASA as initially administered, and one patient was switched to SASP. Among the second-line therapy, the three successful cases consisted of one patient who was switched to a different oral 5-ASA preparation, one patient who underwent desensitization therapy

**Table 2. Comparison of the clinical characteristics of patients in the mesalamine-tolerant and mesalamine-intolerant groups.**

| | Mesalamine-tolerant group (n = 6) | Mesalamine-intolerant group (n = 22) | Odds ratio | 95% CI | *p* value |
|---|---|---|---|---|---|
| Sex (female/male) | 3/3 | 9/13 | 1.444 | 0.236–8.844 | 1.000 |
| Age at diagnosis (years): IQR | 24.5 (21.0–31.0) | 41.5 (32.0–58.0) | | | 0.006 |
| Disease location (%) | | | | | |
| Extensive | 3 (50.0) | 15 (68.2) | 2.143 | 0.342–13.420 | 0.634 |
| Left-sided | 2 (33.3) | 3 (13.6) | 0.316 | 0.039–2.550 | 0.285 |
| Rectum | 1 (16.7) | 4 (18.2) | 1.111 | 0.100–12.308 | 1.000 |
| First oral mesalamine (%) | | | | | |
| Time-dependent release | 1 (16.7) | 7 (31.8) | 2.333 | 0.228–23.908 | 0.640 |
| pH-dependent release | 0 (0) | 6 (27.3) | 1.375 | 1.065–1.776 | 0.289 |
| Multimatrix | 5 (83.3) | 9 (40.9) | 0.138 | 0.014–1.394 | 0.165 |
| Symptoms of adverse events for the first oral mesalamine (%) | | | | | |
| Fever | 4 (66.7) | 13 (59.1) | 0.722 | 0.108–4.820 | 1.000 |
| Diarrhea and bloody stools | 1 (16.7) | 12 (54.5) | 6.000 | 0.598–60.158 | 0.173 |
| Abdominal pain | 2 (33.3) | 5 (22.7) | 0.588 | 0.082–4.212 | 0.622 |
| Joint pain | 1 (16.7) | 3 (13.6) | 0.789 | 0.067–9.317 | 1.000 |
| Nausea | 1 (16.7) | 1 (4.5) | 0.238 | 0.013–4.497 | 0.389 |
| Headache | 0 (0) | 1 (4.5) | 1.048 | 0.956–1.148 | 1.000 |
| Fatigue | 0 (0) | 1 (4.5) | 1.048 | 0.956–1.148 | 1.000 |
| Skin rash | 0 (0) | 1 (4.5) | 1.048 | 0.956–1.148 | 1.000 |
| Pancreatitis | 0 (0) | 1 (4.5) | 1.048 | 0.956–1.148 | 1.000 |
| Pneumonia | 0 (0) | 1 (4.5) | 1.048 | 0.956–1.148 | 1.000 |
| Bone marrow suppression | 0 (0) | 1 (4.5) | 1.048 | 0.956–1.148 | 1.000 |
| Liver dysfunction | 0 (0) | 1 (4.5) | 1.048 | 0.956–1.148 | 1.000 |
| Duration to the onset of adverse events for first oral mesalamine (days); IQR | 12.0 (10.0–27.0) | 13.0 (8.0–20.0) | | | 0.955 |

Values are presented as medians or numbers. 5-ASA, 5-aminosalicylate; IQR, interquartile range.

**Table 3. Comparison of patients who underwent DLST in the mesalamine-tolerant and mesalamine-intolerant groups.**

| | Mesalamine-tolerant group (n = 6) | Mesalamine-intolerant group (n = 22) | Odds ratio | 95% CI | *p* value |
|---|---|---|---|---|---|
| Positive F-DLST (%) | 0 (0) | 13 (59.1) | 2.444 | 1.479–4.039 | 0.018 |
| F-DLST (IQR) | 83.5 (74.0–90.0) | 250.5 (159.0–391.0) | | | 0.002 |
| Positive H-DLST (%) | 2 (33.3) | 18 (81.8) | 9.000 | 1.201–67.417 | 0.038 |
| H-DLST (IQR) | 153.5 (90.0–226.0) | 411.0 (222.0–562.0) | | | 0.005 |
| Positive M-DLST (%) | 0 (0) | 15 (68.1) | 3.143 | 1.705–5.794 | 0.005 |
| M-DLST (IQR) | 110.4 (84.3–118.75) | 311.1 (166.5–384.3) | | | 0.001 |

Values are presented as medians or numbers. DLST, drug-induced lymphocyte stimulation test; F-DLST, DLST for the first oral 5-aminosalicylate; H-DLST, highest DLST among various types of oral mesalamine preparations; IQR, interquartile range; M-DLST, mean DLST among various types of oral mesalamine preparations.

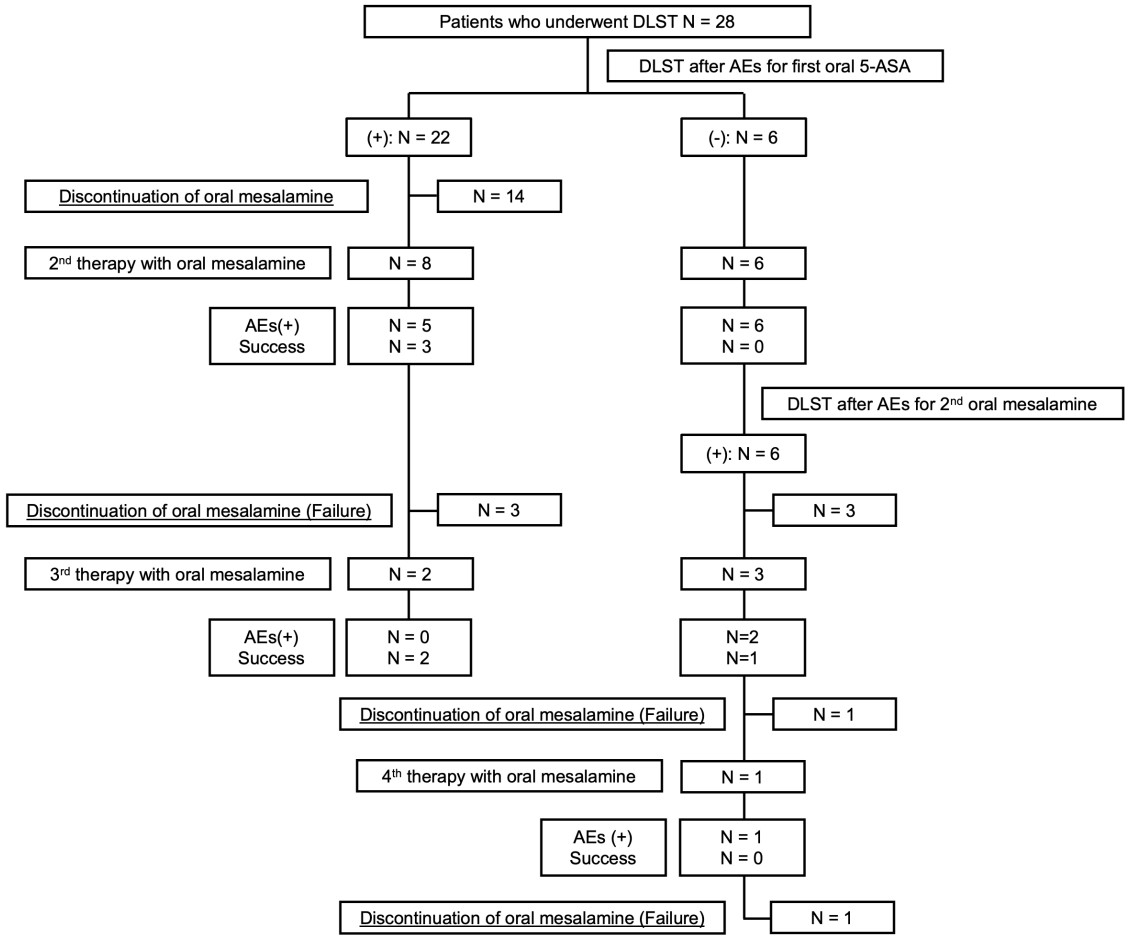

**Fig 2. Flowchart of the resumption of oral mesalamine therapy after adverse events caused by the first oral 5-aminosalicylate.**

with a time-dependent release type of 5-ASA that differed from their initial formulation, and one patient who underwent desensitization therapy using the same type of time-dependent release 5-ASA as initially administered. For the third-line therapy, four patients were switched to SASP, and one patient underwent desensitization therapy with a time-dependent release type of 5-ASA that differed from the initial formulation. Among the third-line therapy, the three successful cases consisted of two patients who were switched to SASP and one patient who underwent desensitization therapy with a time-dependent release type of 5-ASA that differed from the initial formulation. Furthermore, for the fourth-line therapy, one patient was switched to a different type of oral 5-ASA preparation. Desensitization therapy with a time-dependent release type of 5-ASA was initiated at 5 mg/day. Treatment with a different type of oral 5-ASA preparation was started at a median dose of 2,400 (IQR: 800–2,400) mg/day, whereas SASP therapy was started at a median dose of 500 (IQR: 375–1,250) mg/day. All treatment regimens were titrated upward to reach at least 2,000 mg/day. Six patients who achieved successful retreatment maintained oral mesalamine therapy at a minimum dose of 2,000 mg/day for at least 6 months, and none discontinued treatment thereafter. No significant correlation was found between positive DLSTs and retreatment outcomes. Higher F-DLSTs (median 83.5 vs. 168.5, p = 0.033) and M-DLSTs (median 110.4 vs. 172.75, p = 0.010) were associated with failure of oral mesalamine retreatment (Table 4). Additionally, older age at diagnosis was associated with retreatment failure (median 24.5 vs. 39.0, p = 0.033). No significant differences in sex, disease location, symptoms of AEs for the first

**Table 4. Comparison of DLST and failed retreatment with oral mesalamine therapy.**

|  | Success | Failure | Odds ratio | 95% CI | p value |
|---|---|---|---|---|---|
|  | n = 6 | n = 8 |  |  |  |
| Positive F-DLST (%) | 0 (0) | 3 (37.5) | 1.600 | 0.935–2.737 | 0.209 |
| F-DLST; IQR | 83.5 (74.0–90.0) | 168.5 (127.0–605.5) |  |  | 0.033 |
| Positive H-DLST (%) | 2 (33.3) | 5 (62.5) | 3.333 | 0.362–30.701 | 0.592 |
| H-DLST; IQR | 153.5 (90.0–226.0) | 259.5 (168.5–689.5) |  |  | 0.071 |
| Positive M-DLST (%) | 0 (0) | 3 (37.5) | 1.600 | 0.935–2.737 | 0.209 |
| M-DLST; IQR | 110.4 (84.3–118.75) | 172.75 (136.9–518.8) |  |  | 0.010 |

Values are presented as medians or numbers. DLST, drug-induced lymphocyte stimulation test; F-DLST, DLST for the first oral 5-aminosalicylate; H-DLST, highest DLST among various types of oral mesalamine preparations; IQR, interquartile range; M-DLST, mean DLST among various types of oral mesalamine preparations.

oral 5-ASA, or time to onset of AEs for the first oral 5-ASA were detected between the retreatment success and failure groups (Table 5).

The relationship between the number of positive DLSTs for various types of mesalamine and the failure of retreatment with mesalamine is shown in Fig 3. Retreatment failed in 42.9% (three of seven) of patients with no positive DLSTs, in 33.3% (one of three) of patients with one positive DLST, and in 100% (all four) of patients with two or three positive DLSTs.

### Relationship between mesalamine intolerance and DLST cutoff values

The diagnostic potential of the DLST for predicting mesalamine intolerance was evaluated by plotting ROC curves (Fig 4a). A cutoff value of 96.0 for F-DLST yielded a sensitivity of 95.5%, a specificity of 83.3%, and a maximum Youden index (0.788) with an area under the curve (AUC) of 0.928 ($p < 0.001$). A cutoff value of 288.5 for H-DLST resulted in a

**Table 5. Demographic characteristics of patients who received retreatment with oral mesalamine after developing adverse events due to the first mesalamine therapy.**

|  | Success | Failure | Odds ratio | 95% CI | p value |
|---|---|---|---|---|---|
|  | (n = 6) | (n = 8) |  |  |  |
| Sex (female/male) | 3/3 | 2/6 | 3.000 | 0.312–28.841 | 0.580 |
| Age at diagnosis (years): IQR | 24.5 (21.0–31.0) | 39.0 (31.0–45.5) |  |  | 0.033 |
| Disease location (%) |  |  |  |  |  |
| Extensive | 3 (50.0) | 6 (75.0) | 3.000 | 0.312–28.841 | 0.580 |
| Left-sided | 2 (33.3) | 1 (12.5) | 0.286 | 0.019–4.237 | 0.538 |
| Rectum | 1 (16.7) | 1 (12.5) | 0.714 | 0.036–14.347 | 1.000 |
| Symptoms of adverse events for the first oral mesalamine (%) |  |  |  |  |  |
| Fever | 4 (66.7) | 5 (62.5) | 0.833 | 0.090–7.675 | 1.000 |
| Diarrhea and bloody stools | 1 (16.7) | 4 (50.0) | 5.000 | 0.388–64.387 | 0.301 |
| Abdominal pain | 2 (33.3) | 2 (25.0) | 0.667 | 0.065–6.871 | 1.000 |
| Nausea | 1 (16.7) | 1 (12.5) | 0.714 | 0.036–14.347 | 1.000 |
| Headache | 0 (0) | 1 (12.5) | 1.143 | 0.880–1.485 | 1.000 |
| Fatigue | 0 (0) | 1 (12.5) | 1.143 | 0.880–1.485 | 1.000 |
| Joint pain | 1 (16.7) | 3 (37.5) | 3.000 | 0.227–39.608 | 0.580 |
| Time to the onset of adverse events for the first oral mesalamine (days); IQR | 12.0 (10.0–27.0) | 16.0 (4.5–22.0) |  |  | 0.948 |

Values are presented as medians or numbers. IQR, interquartile range.

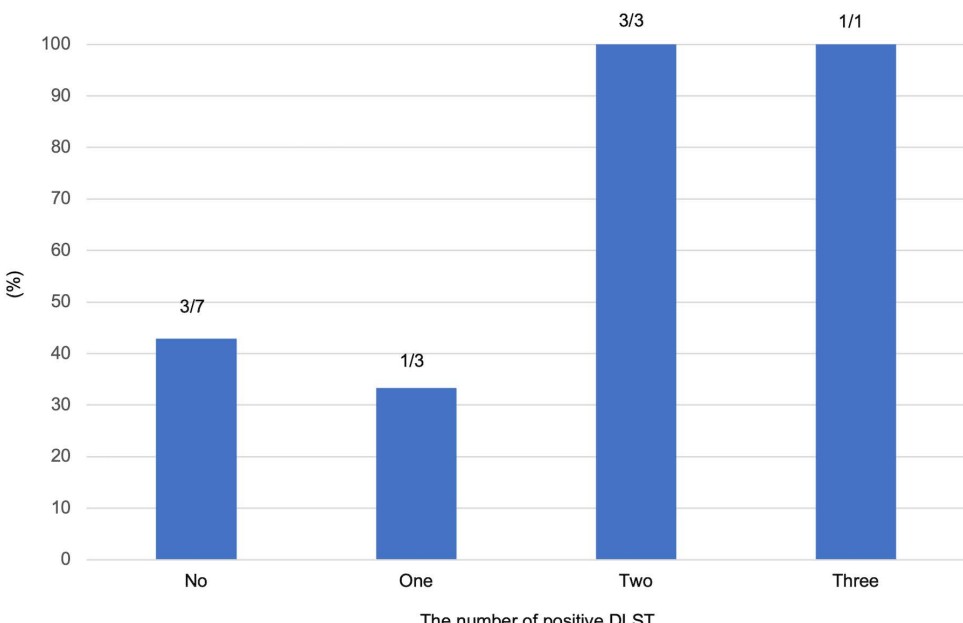

**Fig 3. Relationship between the number of positive DLSTs and mesalamine retreatment failure.**

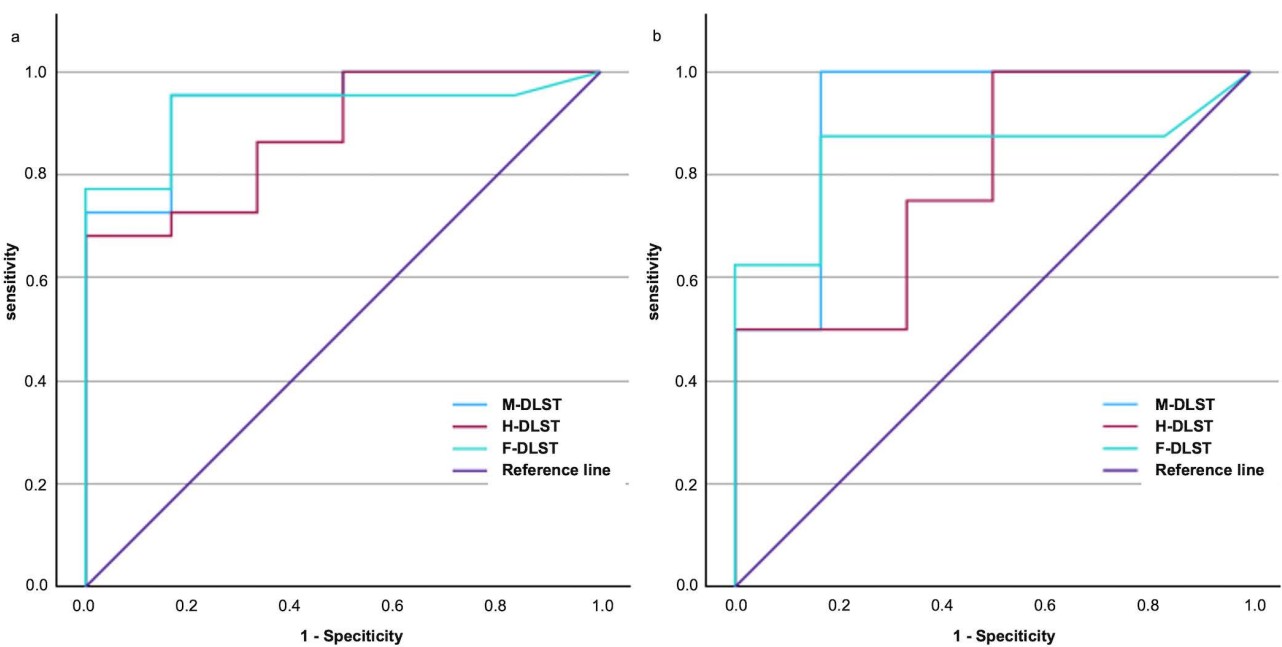

**Fig 4. Receiver operating characteristic curves for DLST.** (a) Association of DLST with the diagnosis of mesalamine intolerance. (b) Relationship between DLST and oral mesalamine retreatment failure.

sensitivity of 68.2%, a specificity of 100%, and a maximum Youden index (0.682) with an AUC of 0.879 (p < 0.001). The cutoff value of 120.2 for M-DLST exhibited a sensitivity of 95.5%, a specificity of 83.3%, and a maximum Youden index (0.788) with an AUC of 0.939 (p < 0.001).

Fig 4b shows the ROC curves for the relationship between DLST and mesalamine retreatment failure. A cutoff value for F-DLST of 98.5 yielded a sensitivity of 87.5%, a specificity of 83.3%, and the maximum Youden index (0.708) was obtained, with an AUC of 0.844 (p = 0.003). The sensitivity and specificity of H-DLST at a cutoff value of 140.0 were 100% and 50.0%, respectively, with a maximal Youden index (0.500) and AUC of 0.792 (p = 0.020). A cutoff value of 120.2 for M-DLST yielded a sensitivity of 100%, a specificity of 83.3%, a maximum Youden index (0.833), and an AUC of 0.917 (P < 0.0001).

## Discussion

In this retrospective single-center cohort study, the incidence of AEs in patients receiving the first oral 5-ASA was 7.2%. Positive F-DLST, H-DLST, and M-DLST were significantly more frequent in the mesalamine-intolerant group compared with positive tests in the mesalamine-tolerant group. The F-DLST, H-DLST, and M-DLST in the mesalamine-intolerant group were higher than the test values in the mesalamine-tolerant group, confirming the usefulness of the DLST in diagnosing mesalamine intolerance. In patients retreated with mesalamine, higher F-DLST, and M-DLST were associated with retreatment failure. The AUCs for the ROC curves of F-DLST, H-DLST, and M-DLST were high, confirming the usefulness of the DLST in diagnosing mesalamine intolerance and predicting retreatment failure with mesalamine. To the best of our knowledge, this is the first study to evaluate the use of DLST for various oral mesalamine formulations for diagnosing mesalamine intolerance and failure of mesalamine retreatment in patients with UC.

Mesalamine intolerance is still not clearly defined [28]. In this study, we used the definition of mesalamine intolerance from a multicenter study reported in Japan [13]. Mesalamine intolerance is often defined as the presence of AEs due to at least one mesalamine formulation, independent of whether oral mesalamine can be continued [12,24–26]. Patients with mesalamine intolerance often use advanced therapies such as immunomodulators and biologics [13], which also carry the risk of AEs [29,30] and malignancy [31]. Therefore, continuing mesalamine therapy whenever possible is important in avoiding unnecessary advanced therapies and reducing the risk of colectomy [4]. However, mesalamine retreatment can cause major AEs; thus, it should be used with caution [16,26]. The precise mechanisms underlying mesalamine intolerance remain unclear; however, immune-mediated reactions to 5-ASA, sulfapyridine, or formulation excipients were suggested [28]. A genome-wide association study revealed a significant genetic predisposition to mesalamine intolerance. Moreover, it identified rs144384547 as a genome-wide significant susceptibility single-nucleotide polymorphism associated with mesalamine-induced fever and diarrhea [32]. The mesalamine-intolerant group had higher abundance levels of *Faecalibacterium*, *Streptococcus*, and *Clostridium* than the mesalamine-tolerant group. Based on this finding, dysbiosis may contribute to the development of mesalamine intolerance [33]. DLST, which is commonly used to assess T-cell sensitization to drugs, can evaluate lymphocyte proliferation by determining the incorporation of $^3$H-thymidine into the DNA of proliferating lymphocytes. This assay is based on the principle that antigen-specific T cells undergo clonal expansion upon antigen recognition [34]. Accordingly, if mesalamine intolerance involves an immunologically mediated hypersensitivity reaction, a high SI would be expected. The mesalamine retreatment rate ranges from 22.6% to 63.9% [12,24–26], and our retreatment rate (50.0%) was comparable to these values. This high retreatment rate indicates that our hospital has an aggressive retreatment policy, considering the degree of side effects and DLST results.

Significant differences in positive F-DLST, H-DLST, and M-DLST and high F-DLST, H-DLST, and M-DLST were identified. An association between mesalamine intolerance as defined in this study and DLST, indicating the usefulness of DLST, has not been previously reported. However, predictors of mesalamine intolerance have been previously reported. In addition to the DLST results, an older age at diagnosis was significantly associated with mesalamine intolerance and failed retreatment with mesalamine. To the best of our knowledge, previous studies have not reported a similar

association between mesalamine intolerance and age at diagnosis. Hiraoka et al. reported that successful retreatment with mesalamine was associated with the lack of fever and abdominal pain (odds ratio = 4.64; 95% confidence interval, 0.85–25.3, p = 0.052) [12]. Suzuki et al. reported that fever and diarrhea caused by mesalamine allergy were associated with rs144384547 located upstream of RGS17. The frequency of mesalamine-induced fever and diarrhea was higher in patients heterozygous for rs144384547 compared with patients without the risk allele for this SNP (22.0% vs. 2.34%) [32]. In our study, fever was also the most common AE of 5-ASA but was not a predictor of mesalamine intolerance. The incidence of fever was low (22.0%) in the Suzuki study, even in patients with the gene mutation. Shuji et al. reported that pancolitis tended to be associated with mesalamine intolerance (p = 0.059) [13]. In our study, no significant differences in disease location were detected between patients with and without mesalamine intolerance. Mesalamine intolerance can occur in any disease location.

The mesalamine retreatment success rate in patients who experienced an AE due to the first oral 5-ASA was 42.9% in our study. Significant differences in the success of mesalamine retreatment were detected between patients with high and low mesalamine F-DLST and M-DLST. Previous studies reported mesalamine retreatment success rates of 0%–85.7% [12,24–26], and some reports indicated that positive DLST results to the suspect drug did not affect retreatment [24,25]. Desensitization therapy with time-dependent 5-ASA is often successful, even in patients with positive DLSTs [25]. However, Hirotaka et al. reported that retreatment of five DLST-positive patients with mesalamine resulted in AEs in all patients [26]. The differences in results may be due to allergies, the retreatment starting dose, the influence of excipients, and the involvement of the rs144384547 gene. No association between positive DLSTs and retreatment was detected, but F-DLST and M-DLST were higher in the retreatment failure group compared with the success group, suggesting that the cutoff values for retreatment may be different. All patients with positive DLSTs for more than one mesalamine failed retreatment. Positive DLSTs to more than one mesalamine may be due to allergy to mesalamine and should be carefully considered before retreatment.

DLST cutoff values for mesalamine intolerance or retreatment have not been reported. The cutoff values of the DLST vary for different drugs [21,35]. The reported sensitivity of DLST when a 5-ASA caused AEs is 0.24, the specificity is 0.81, the false positive rate is 0.195, and the false negative rate is 0.76 [21]. In the present study, F-DLST, H-DLST, and M-DLST had high AUCs for predicting mesalamine intolerance; the cutoff values for F-DLST and M-DLST were less than 180%, and the cutoff value for H-DLST was 288.5%. The cutoff values for F-DLST, H-DLST, and M-DLST for predicting retreatment failure were below 180% and correlated well. Importantly, all patients with two or more positive DLSTs failed retreatment, indicating a high likelihood of true mesalamine allergy. In contrast, patients with negative or only one positive DLST often tolerated retreatment, suggesting that intolerance may be formulation-specific rather than related to mesalamine itself. These findings provide a preliminary framework: DLST should be performed after the initial onset of intolerance symptoms following the first oral 5-ASA, and it is advisable to test all available oral mesalamine formulations. Retreatment should be avoided when multiple DLSTs are positive, whereas cautious reintroduction may be considered in cases with negative or single-positive results.

This study has several limitations. First, this was a single-center, retrospective cohort study with a limited and uneven sample size, which restricts the statistical power and limits the external validity and generalizability of the findings. Moreover, only univariate comparisons were performed, and potential confounders such as age, disease extent, type of first 5-ASA, 5-ASA dose, severity of AE, and corticosteroid use were not adjusted for. Although a multivariable logistic regression would be essential to clarify independent predictors of mesalamine intolerance and retreatment outcomes, this was not feasible owing to the small number of patients. Future studies with larger sample sizes are needed to address these potential confounders in a more robust manner. Second, treatment decisions, including mesalamine retreatment and the performance of DLSTs, were made at the discretion of the attending physicians, which may have introduced variability. Additionally, although the DLST showed significant associations with mesalamine intolerance, its clinical applicability remains uncertain due to limited sensitivity and specificity as well as the risk of both false positives and false negatives.

Our findings suggest a preliminary framework for clinical decision-making; however, DLST should not yet be applied in routine practice, and further validation in larger, multicenter prospective cohorts is required. Third, the ROC-derived cutoff values, odds ratios, and AUCs were derived and evaluated within the same small dataset, raising concerns about statistical overfitting and overly optimistic estimates. Therefore, external validation in larger, multicenter prospective cohorts will be necessary to confirm the diagnostic performance of the DLST and the generalizability of these thresholds. Finally, to ensure diagnostic clarity and sufficient follow-up, patients with incomplete data, those who had used topical mesalamine preparations before oral mesalamine formulations, and those with a short history of hospital visits were excluded from the analysis. However, these exclusion criteria might have introduced selection bias. In conclusion, future multicenter prospective cohort studies are essential to validate the diagnostic value of the DLST and the clinical management of mesalamine intolerance.

## Conclusions

DLSTs for various oral mesalamine formulations may have a predictive value in diagnosing mesalamine intolerance and assessing the likelihood of successful retreatment. Nevertheless, their clinical application requires caution owing to potential false-positive and false-negative results. To improve diagnostic accuracy, DLSTs should be interpreted along with clinical symptoms and, if available, other emerging diagnostic modalities such as genetic testing, once clinically validated.

## Acknowledgments

The authors thank *Enago* (www.enago.jp) for the English language review.

## Author contributions

**Conceptualization:** Akira Madarame, Tadashi Ichimiya.

**Data curation:** Akira Madarame, Kumiko Uchida, Tadashi Ichimiya, Sakiko Naito, Yoshiya Yamauchi, Takashi Morise, Yasuyuki Kagawa, Takahiro Muramatsu.

**Formal analysis:** Akira Madarame.

**Methodology:** Akira Madarame, Masakatsu Fukuzawa, Takao Itoi.

**Project administration:** Akira Madarame.

**Visualization:** Akira Madarame.

**Writing – original draft:** Akira Madarame.

**Writing – review & editing:** Masakatsu Fukuzawa, Kumiko Uchida, Tadashi Ichimiya, Sakiko Naito, Yoshiya Yamauchi, Takashi Morise, Yasuyuki Kagawa, Takahiro Muramatsu, Takao Itoi.

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
