## [Decision Letter · Decision Letter 0]

9 May 2025

Dear Dr. Madarame,

Thank you for submitting your manuscript to PLOS ONE. After careful consideration, we feel that it has merit but does not fully meet PLOS ONE’s publication criteria as it currently stands. Therefore, we invite you to submit a revised version of the manuscript that addresses the points raised during the review process.

We look forward to receiving your revised manuscript.

Kind regards,

Dr. Mohammed Misbah Ul Haq, Pharm-D

Academic Editor

PLOS ONE

Journal Requirements:

2.  Please remove all personal information, ensure that the data shared are in accordance with participant consent, and re-upload a fully anonymized data set.

Reviewers' comments:

Reviewer's Responses to Questions

**Comments to the Author**

1. Is the manuscript technically sound, and do the data support the conclusions?

Reviewer #1: No

Reviewer #2: Yes

2. Has the statistical analysis been performed appropriately and rigorously?

Reviewer #1: I Don't Know

Reviewer #2: Yes

3. Have the authors made all data underlying the findings in their manuscript fully available?

Reviewer #1: No

Reviewer #2: Yes

4. Is the manuscript presented in an intelligible fashion and written in standard English?

Reviewer #1: Yes

Reviewer #2: Yes

Reviewer #1: Significance of the drug-induced lymphocyte stimulation test for various oral mesalamines in ulcerative colitis with mesalamine intolerance

This is an interesting topic.

However, the clinical usefulness and reliability of DLST testing for mesalazine is very low. For this reason, DLST testing is being performed less and less. As a result, it is difficult to find meaning in research discussing the usefulness of DLST testing. There have already been several papers reporting that mesalazine intolerance has a poor prognosis, so this study lacks novelty. It is also unfortunate that the number of studies is small.

Reviewer #2: 1.As a single-center retrospective study, the limited sample size (n=28) may result in insufficient statistical power. Expanding the cohort or conducting multicenter collaborations would improve the reliability of the findings. 2.The exclusion of certain cases (e.g., patients receiving topical mesalazine) during screening could introduce selection bias. The potential impact of this limitation should be explicitly addressed in the Discussion section. 3.The manuscript lacks detailed methodological descriptions of the drug lymphocyte stimulation test (DLST), including critical parameters such as drug concentrations and incubation periods. Supplementary procedural details are recommended to ensure experimental reproducibility. 4.The immunological mechanisms of DLST positivity and mesalazine intolerance are not discussed in depth (e.g., whether it is an allergic reaction or non-immunotoxic). It is recommended that additional mechanistic hypotheses be added to the existing literature. 5. The conclusions emphasise the predictive value of DLST, but there may be a problem of false positives/negatives in practical application. Discussion is needed on how to improve diagnostic accuracy in combination with clinical symptoms or other tests (e.g., genetic testing).

**Do you want your identity to be public for this peer review?** For information about this choice, including consent withdrawal, please see our Privacy Policy

Reviewer #1: No

Reviewer #2: No

---

## [Author Response · Author response to Decision Letter 1]

29 May 2025

Manuscript ID: PONE-D-25-17000

Title: Significance of the drug-induced lymphocyte stimulation test for various oral mesalamines in ulcerative colitis with mesalamine intolerance

Dear Editors and Reviewers,

We would like to thank you for the opportunity to revise our manuscript and the insightful comments. We have carefully addressed each point raised by the reviewers and have revised the manuscript accordingly. Our detailed responses to each comment are provided below.

Reviewer #1

Comment: The clinical usefulness and reliability of DLST testing for mesalazine is very low. For this reason, DLST testing is being performed less and less. As a result, it is difficult to find meaning in research discussing the usefulness of DLST testing. There have already been several papers reporting that mesalazine intolerance has a poor prognosis, so this study lacks novelty.

Response:

Given its relatively low sensitivity and specificity, the use of DLST in clinical practice is limited. Further, its application in other settings has decreased. However, in Japan, DLST is still used as a supportive tool for evaluating suspected drug-induced hypersensitivity reactions, such as mesalamine intolerance.

This study differs from previous ones because previous studies commonly assessed the DLST results only for mesalamine preparation that were initially suspected. Moreover, the present study examined the DLST results for various types of oral mesalamine in each patient, thereby allowing the evaluation of whether a higher M-DLST across various formulations is associated with clinical intolerance and retreatment failure.

In addition, this study explored an optimal SI cutoff value for predicting intolerance, which has not been clearly addressed in previous studies. We believe that our findings offer novel contributions to the field, particularly in using DLST more effectively as part of a multifaceted diagnostic approach. These points have been added to the Introduction and Discussion of the revised manuscript.

Comment: The number of cases is small.

Response:

We acknowledge that this single-center retrospective study was limited by the small sample size. This has been explicitly discussed in the revised Discussion.

Reviewer #2

Comment 1: The limited sample size (n=28) may result in insufficient statistical power. Expanding the cohort or conducting multicenter collaborations would improve the reliability of the findings.

Response:

We agree with this limitation, and this has been clearly stated in the Discussion. We also recommended the need to conduct multicenter prospective studies to validate our findings.

Comment 2: The exclusion of certain cases (e.g., patients receiving topical mesalazine) during screening could introduce selection bias. The potential impact of this limitation should be explicitly addressed in the Discussion section.

Response:

Thank you for this valuable suggestion. A statement acknowledging the possibility of selection bias due to our exclusion criteria and its potential effect on the generalizability of the results has been added to the Discussion.

Comment 3: The manuscript lacks detailed methodological descriptions of the DLST, including drug concentrations and incubation periods. Supplementary procedural details are recommended.

Response:

The Methods was revised to include data on drug concentrations, incubation conditions, and control settings used in DLST.

Comment 4: The immunological mechanisms of DLST positivity and mesalamine intolerance are not discussed in depth (e.g., whether it is an allergic reaction or non-immunotoxic).

Response:

Thank you for your valuable comment. As suggested, the Discussion has been revised. In particular, data on the possible immunological mechanisms underlying mesalamine intolerance and DLST positivity were added. Moreover, information on immune-mediated reactions to 5-ASA, sulfapyridine, or formulation excipients, a genome-wide significant SNP (rs144384547) associated with mesalamine-induced symptoms, and possible involvement of dysbiosis was included. Furthermore, an explanation of the DLST principle was added, thereby emphasizing its relevance in detecting T-cell-mediated hypersensitivity, which may be reflected by a high stimulation index in mesalamine-intolerant cases.

Comment 5: The conclusions emphasise the predictive value of DLST, but there may be a problem of false positives/negatives in practical application. Discussion is needed on how to improve diagnostic accuracy.

Response:

The Conclusion and Discussion were revised to support our claims regarding predictive utility. The risks of false-positive and false-negative results were explicitly discussed. Furthermore, we proposed interpreting the DLST findings in consideration of clinical symptoms and, if available, genetic or immunologic markers.

Additional Revisions

We appreciate the editorial request regarding data sharing. As described in the manuscript, the data were anonymized before analysis in compliance with ethical guidelines and were collected using an opt-out procedure approved by our institutional ethics committee. The requirement for informed consent was waived owing to the retrospective nature of the study and the use of anonymized data.

However, after re-examining the dataset in light of journal publication standards, certain combinations of clinical variables (e.g., age, sex, disease characteristics, and treatment responses) could potentially allow the reidentification of individuals if shared externally. Furthermore, participants did not provide explicit consent for external data sharing. According to Japanese law, disclosing potentially identifiable clinical information to parties outside the institution without consent may constitute a legal violation. Therefore, we have removed the Supporting Information and clarified this limitation in the revised Data availability section.

The revised manuscript with track changes, a clean version of the manuscript, and a response letter were uploaded separately.

We confirmed that our protocols do not require registration in protocols.io, as they do not involve novel laboratory methods.

We hope that the revisions and clarifications satisfactorily address the concerns raised. We greatly appreciate the reviewers’ constructive feedback and your consideration of our revised manuscript.

Sincerely,

Akira Madarame, MD

---

## [Decision Letter · Decision Letter 1]

4 Jul 2025

Dear Dr. Madarame,

We look forward to receiving your revised manuscript.

Kind regards,

Dr. Mohammed Misbah Ul Haq, Pharm-D

Academic Editor

PLOS ONE

Reviewers' comments:

Reviewer's Responses to Questions

**Comments to the Author**

Reviewer #3: All comments have been addressed

Reviewer #4: (No Response)

2. Is the manuscript technically sound, and do the data support the conclusions?

Reviewer #3: Partly

Reviewer #4: (No Response)

3. Has the statistical analysis been performed appropriately and rigorously?

Reviewer #3: Yes

Reviewer #4: (No Response)

4. Have the authors made all data underlying the findings in their manuscript fully available?

Reviewer #3: No

Reviewer #4: (No Response)

5. Is the manuscript presented in an intelligible fashion and written in standard English?

Reviewer #3: Yes

Reviewer #4: (No Response)

Reviewer #3: This manuscript suggests that DLSTs are useful for retreatment mesalamine in cases with side effects for first 5-ASA.

However, it requires some revisions as listed below.

1. The authors divided the 28 patients who experienced AEs from the first 5-ASA treatment into two groups: a tolerable group of six and an intolerable group of 22. Was this determined by retreatment after discontinuing the first 5-ASA to assess tolerability or intolerance (Page 6, line 107-111)? The Results section states 14 patients underwent retreatment. The study flow and Methods section needs to be clearly described.

2. The definitions of success and failure for 5-ASA retreatment should be indicated. Fig 2 shows that mesalamine retreatment was undergone multiple times. However, it is unclear at what point the retreatment was deemed a success or failure.

3. The authors should provide information on mesalamine retreatment details that could affect outcomes, such as dosage and the presence or absence of desensitization therapy.

4. The spelling of mesalamine should be consistent throughout the text and figures.

Reviewer #4: (No Response)

**Do you want your identity to be public for this peer review?** For information about this choice, including consent withdrawal, please see our Privacy Policy

Reviewer #3: No

Reviewer #4: No

---

## [Author Response · Author response to Decision Letter 2]

20 Jul 2025

Manuscript ID: PONE-D-25-17000

Title: Significance of the drug-induced lymphocyte stimulation test for various oral mesalamines in ulcerative colitis with mesalamine intolerance

Dear Editors and Reviewers:

We would like to thank you for the opportunity to revise our manuscript as well as for the insightful comments. We have carefully addressed each point raised by the reviewers and have revised the manuscript accordingly. Our detailed responses to each comment are provided below.

Reviewer #3

Comment 1: The authors divided the 28 patients who experienced AEs from the first 5-ASA treatment into two groups: a tolerable group of six and an intolerable group of 22. Was this determined by retreatment after discontinuing the first 5-ASA to assess tolerability or intolerance (Page 6, line 107-111)? The Results section states 14 patients underwent retreatment. The study flow and Methods section needs to be clearly described.

Response:

Thank you for your valuable comment. We apologize for the lack of clarity in the description regarding the patient grouping. In the revised manuscript, we have clarified that mesalamine intolerance was defined as either (1) the occurrence of adverse events (AEs) within 6 months of the first oral 5-ASA and the inability to continue taking oral mesalamines owing to intolerance upon retreatment or (2) patients whose initial AEs were severe or involved significant organ toxicity, and therefore, did not receive mesalamine retreatment. We have added these definitions and clarifications to the Methods section, under the heading Data collection and definitions (Page 6, lines 119–124).

Comment 2: The definitions of success and failure for 5-ASA retreatment should be indicated. Fig 2 shows that mesalamine retreatment was undergone multiple times. However, it is unclear at what point the retreatment was deemed a success or failure.

Response:

Thank you for bringing this to our attention. We have added clear definitions of “retreatment success” and “failure” to the Methods section, under the heading Data collection and definitions. Retreatment success was defined as the ability to continue taking at least one oral mesalamine formulation for >6 months after starting the first oral 5-ASA without AEs, and retreatment failure was defined as discontinuation of mesalamine retreatment owing to AEs (Page 6, lines 126–129).

To make this clearer, we have revised the description of Fig. 2 as well.

Comment 3: The authors should provide information on mesalamine retreatment details that could affect outcomes, such as dosage and the presence or absence of desensitization therapy.

Response:

Thank you for your insightful comment. We have added detailed information in the Results section on the methods of mesalamine retreatment that could affect outcomes. Specifically, we describe the three approaches used, which are as follows: (1) switching to a different type of oral 5-ASA, (2) performing desensitization therapy, and (3) switching to SASP.

Additionally, the initial dose at retreatment was specified, and all treatment regimens were titrated upward to reach at least 2,000 mg/day (Page 14, lines 248–261).

Comment 4: The spelling of mesalamine should be consistent throughout the text and figures.

Response:

Thank you for bringing this to our attention. Accordingly, we have carefully checked the entire manuscript, including the text and figures, to ensure that the spelling of mesalamine is consistent throughout.

Major comments

Materials and methods

Comment 1: Why were patients excluded with previous topical 5-ASA drugs? It is a potential confounder, and excluding them results in selection bias.

Response: Thank you for your insightful comment. We acknowledge the reports regarding mesalamine intolerance caused by topical 5-ASA drugs. However, this study specifically aimed to investigate mesalamine intolerance and retreatment outcomes in patients who experienced AEs from their first dose of oral 5-ASA formulation. The inclusion of patients who only experienced intolerance to topical 5-ASA preparations may introduce heterogeneity as their pharmacokinetics and routes of exposure are different from those of oral formulations. Therefore, patients who previously used topical 5-ASA were excluded to maintain the consistency and validity of the study cohort. For clarity, we have added this rationale to the Materials and Methods section (Page 5, lines 102–108).

Comment 2: It should be emphasized that reintroduction of a different type of 5-ASA formulation is only acceptable and ethical in cases of mild intolerance. In cases of severe adverse events, such as severe renal failure or pancreatitis, re-exposure should be strictly avoided. This important distinction should be clearly stated.

Response: Thank you for this important comment. We agree that retreatment with a different type of 5-ASA formulation should only be considered in cases of mild intolerance without major organ damage.

To clarify this ethical consideration, we have added the following sentence to the Materials and Methods section: “Patients who experienced only mild AEs from the first oral 5-ASA and had no organ damage received mesalamine retreatment after consultation with their attending physician.” (Page 6, lines 124–126)

Results section

Comment 1: Very small and highly imbalanced sample. The comparison between only six and twenty-two patients—set against an already limited total sample size—substantially restricts statistical power and calls into question the external validity and generalizability of the study’s conclusions.

Insufficient statistical modelling. Only univariate comparisons were performed; potential confounders (age, disease extent, 5-ASA dose, AE severity, corticosteroid use) were not adjusted for. A multivariable logistic regression is essential to clarify independent predictors, however, due to the small sample size, it may not be possible.

Response: Thank you for pointing out this important limitation. We completely agree that the small and uneven sample size restricts the statistical power and limits the external validity and generalizability of our study findings.

In addition, only univariate analyses were performed, and adjusting for potential confounders such as age, disease extent, 5-ASA dose, AE severity, and corticosteroid use would have strengthened the conclusions. However, considering the limited number of patients enrolled in this retrospective single-center cohort, conducting a robust multivariable logistic regression analysis was not feasible.

To address this concern, we have clearly discussed these limitations in the Discussion section and have emphasized that future studies should validate our findings in larger, multicenter prospective cohorts, with adequate sample sizes, to allow for multivariable adjustment (Page 21–22, lines 396–416).

Comment 2: Why was 5-ASA reinducted in case of myelosuppression? It should be stated, as current ECCO guidelines advise against 5-ASA re-challenge after serious organ toxicity. Was the pancreatitis mild?

Response: Thank you for raising this important point. We agree that mesalamine reintroduction should be strictly avoided in patients with significant organ toxicity, such as myelosuppression or severe pancreatitis, in line with current ECCO guidelines.

In this study, no patients with serious organ toxicity received mesalamine retreatment. We believe that this misunderstanding may have arisen from the original wording of our definition of mesalamine intolerance.

To clarify this ethical consideration, we have revised the definition in the Materials and Methods section to explicitly state that retreatment was only performed in cases without significant organ damage (Page 6, lines 124–126).

Comment 3: Although both F-DLST and M-DLST showed statistically significant associations with mesalamine intolerance, their limited sensitivity and specificity substantially restrict their clinical utility.

The ROC-derived cut-offs yield apparently excellent AUC values; however, because they were optimised and evaluated on the same, very small dataset (n = 28, with only six tolerant cases), the results are likely over-optimistic and may not generalise. External validation in a larger cohort is essential before these thresholds can be recommended for clinical use.

Response: Thank you for this valuable comment. We agree that despite the significant associations of F-DLST and M-DLST with mesalamine intolerance, their limited sensitivity and specificity substantially restrict their current clinical utility.

We also acknowledge that the ROC-derived cutoff values and AUCs may appear overly optimistic because they were derived and evaluated within the same small cohort (n = 28), which included only six tolerant cases.

To address this, we have further discussed these limitations in the Discussion section and have emphasized that external validation using larger, multicenter prospective cohorts is essential before these cutoff values can be recommended for routine clinical practice (Page 22, lines 405–410).

Minor comments

Introduction

Comment 1: The phrase ‘Salazosulfapyridine (SASP), one of the first 5-ASAs’ is misleading. SASP is a sulfonamide-based prodrug that delivers 5-ASA after bacterial azo-reduction in the colon. Because of its sulfapyridine moiety, SASP differs chemically, pharmacokinetically, and in its adverse-event profile from the later, pure mesalazine formulations. Please rephrase it.

Response: Thank you for raising this point. We agree that the original phrasing may have been misleading. As suggested, we have revised the sentence to clarify that SASP is a sulfonamide-based prodrug that delivers 5-ASA to the colon via bacterial azo-reduction and that it differs chemically, pharmacokinetically, and in its adverse event profile from the later, pure mesalamine formulations (Page 3, lines 45–49).

Materials and methods

Comment 1: As Salazopyrin is not a classical 5-ASA and , I would not include it in the study as it can increase bias. Both its pharmacokinetics and immunologic toxicity differ substantially from those of single-agent 5-ASA preparations. However, as I can see it in the methods section, no patients received Salazopyrin among the observed 28 patients.

Response: Thank you for your comment. We agree that the inclusion of patients who received SASP could introduce bias because SASP has substantially different pharmacokinetics and immunologic toxicity from single-agent 5-ASA formulations.

To avoid this potential bias, we have explicitly stated in the Materials and Methods that patients who initiated oral SASP at diagnosis were excluded from the study (Page 5, line 99–100).

The revised manuscript with tracked changes, a clean version of the manuscript, and a response letter were uploaded separately.

We confirmed that our protocols do not require registration in protocols.io because they do not involve novel laboratory methods.

We hope that the revisions and clarifications satisfactorily address the concerns of the reviewers. We greatly appreciate the constructive feedback of the reviewers and your consideration of our revised manuscript. We hope that the revised manuscript will be acceptable for publication in PLOS ONE.

Sincerely,

Akira Madarame, MD

---

## [Decision Letter · Decision Letter 2]

1 Sep 2025

Dear Dr. Madarame,

We look forward to receiving your revised manuscript.

Kind regards,

Dr. Mohammed Misbah Ul Haq, Pharm-D

Academic Editor

PLOS ONE

Journal Requirements:

Additional Editor Comment:

Reviewer #3:

Reviewer #5:

Reviewers' comments:

Reviewer's Responses to Questions

**Comments to the Author**

Reviewer #3: All comments have been addressed

Reviewer #5: All comments have been addressed

2. Is the manuscript technically sound, and do the data support the conclusions?

Reviewer #3: Partly

Reviewer #5: Partly

3. Has the statistical analysis been performed appropriately and rigorously?

Reviewer #3: Yes

Reviewer #5: Yes

4. Have the authors made all data underlying the findings in their manuscript fully available?

Reviewer #3: No

Reviewer #5: Yes

5. Is the manuscript presented in an intelligible fashion and written in standard English?

Reviewer #3: Yes

Reviewer #5: Yes

Reviewer #3: Comments to the Author

The authors has responded properly to the reviewer's comments, so I have no additional comments.

Reviewer #5: This manuscript investigates the diagnostic and predictive value of the drug-induced lymphocyte stimulation test (DLST) in patients with ulcerative colitis who experienced mesalamine intolerance. The topic is clinically relevant, and the authors provide novel insights by analyzing DLST results for multiple mesalamine formulations. The study is clearly written and addresses an important clinical problem. However, there are some methodological and interpretative issues that need further clarification.

Substantial concern is that sample size is too small.

Statistical limitations:The study is based on a small, single-center cohort (n=28), which limits statistical power. The cutoff values derived from ROC analysis and the odds ratios may be subject to overfitting. The authors should acknowledge this more explicitly and emphasize the need for validation in larger, multicenter prospective studies.

Clinical applicability of DLST:While the results suggest that DLST could be useful in predicting intolerance and retreatment outcomes, its clinical application remains unclear due to potential false positives and negatives. The discussion would be strengthened by presenting specific clinical scenarios (e.g., when DLST should be considered in practice, or how to act upon multiple positive results) and by proposing a preliminary decision-making algorithm.

Adjustment for confounding factors:The analysis relies only on univariate comparisons. Potential confounders, such as age, type of first 5-ASA, severity of adverse events, and corticosteroid use, may influence the results. Although the limited sample size may preclude multivariable analysis, the authors should discuss this limitation and outline how future research could address it.

**Do you want your identity to be public for this peer review?** For information about this choice, including consent withdrawal, please see our Privacy Policy

Reviewer #3: No

Reviewer #5: **Yes: ** Osamu Handa

---

## [Author Response · Author response to Decision Letter 3]

3 Sep 2025

Manuscript ID: PONE-D-25-17000R2

Title: Significance of the drug-induced lymphocyte stimulation test for various oral mesalamines in ulcerative colitis with mesalamine intolerance

Dear Editors and Reviewers:

We sincerely thank the Academic Editor and Reviewers for their thoughtful and constructive comments, which have been very helpful in improving our manuscript. We have carefully revised the manuscript in accordance with the suggestions, and the specific responses to each comment are detailed below. Page and line numbers refer to the revised version of the manuscript.

Reviewer #3

Comment: The authors have responded properly to the reviewer's comments, so I have no additional comments.

Response:

We thank the reviewer for the positive evaluation and for acknowledging our previous revisions. We are grateful for the constructive feedback provided during the review process.

Reviewer #5

Comment (Statistical limitations):

The study is based on a small, single-center cohort (n=28), which limits statistical power. The cutoff values derived from ROC analysis and the odds ratios may be subject to overfitting. The authors should acknowledge this more explicitly and emphasize the need for validation in larger, multicenter prospective studies.

Response:

We thank the reviewer for raising this important point. We fully agree that the small, single-center cohort limits the statistical power and may increase the risk of overfitting for ROC-derived cutoff values and odds ratios. In the revised Discussion, we have explicitly acknowledged these limitations and emphasized the need for validation in larger, multicenter prospective studies to confirm the diagnostic performance and clinical applicability of DLST (page 22, lines 413–417).

Comment (Clinical applicability of DLST):

While the results suggest that DLST could be useful in predicting intolerance and retreatment outcomes, its clinical application remains unclear due to potential false positives and negatives. The discussion would be strengthened by presenting specific clinical scenarios (e.g., when DLST should be considered in practice, or how to act upon multiple positive results) and by proposing a preliminary decision-making algorithm.

Response:

We thank the reviewer for this constructive suggestion. In the revised Discussion, we have added specific descriptions of how DLST may be applied in clinical practice. Based on our findings, we suggest that DLST should be performed after the first onset of intolerance symptoms to oral 5-ASA and ideally include all available mesalamine formulations. We further describe a preliminary framework in which retreatment should be avoided when two or more DLSTs are positive, whereas cautious reintroduction may be considered in patients with negative or single-positive results (page 21, lines 388–396). We have also emphasized that this framework is preliminary and requires validation in larger, multicenter prospective studies before being implemented in routine clinical practice (page 22, lines 410–413).

Comment (Adjustment for confounding factors):

The analysis relies only on univariate comparisons. Potential confounders, such as age, type of first 5-ASA, severity of adverse events, and corticosteroid use, may influence the results. Although the limited sample size may preclude multivariable analysis, the authors should discuss this limitation and outline how future research could address it.

Response:

We thank the reviewer for highlighting this important issue. As suggested, we have added a statement in the Discussion acknowledging that our analysis was limited to univariate comparisons and did not adjust for potential confounders, including age, type of first 5-ASA, severity of adverse events, and corticosteroid use. We have also noted that, although multivariable analysis was not feasible due to the small sample size, future studies with larger cohorts will be necessary to incorporate these factors and clarify independent predictors of mesalamine intolerance and retreatment outcomes (page 21-22, lines 397–405).

We carefully reviewed the reviewer comments and confirm that no additional references were specifically recommended in this revision round. Accordingly, no new citations were added.

The revised manuscript with tracked changes, a clean version of the manuscript, and a response letter were uploaded separately.

We confirmed that our protocols do not require registration in protocols.io because they do not involve novel laboratory methods.

We hope that the revisions and clarifications satisfactorily address the concerns of the reviewers. We greatly appreciate the constructive feedback of the reviewers and your consideration of our revised manuscript. We hope that the revised manuscript will be acceptable for publication in PLOS ONE.

Sincerely,

Akira Madarame, MD

---

## [Decision Letter · Decision Letter 3]

24 Sep 2025

Dear Dr. Madarame,

Thank you for submitting your manuscript to PLOS ONE. After careful consideration, we feel that it has merit but does not fully meet PLOS ONE’s publication criteria as it currently stands. Therefore, we invite you to submit a revised version of the manuscript that addresses the points raised during the review process.

We look forward to receiving your revised manuscript.

Kind regards,

Dr. Mohammed Misbah Ul Haq, Pharm-D

Academic Editor

PLOS ONE

Journal Requirements:

Reviewers' comments:

Reviewer's Responses to Questions

**Comments to the Author**

Reviewer #6: All comments have been addressed

Reviewer #7: (No Response)

2. Is the manuscript technically sound, and do the data support the conclusions?

Reviewer #6: (No Response)

Reviewer #7: No

3. Has the statistical analysis been performed appropriately and rigorously?

Reviewer #6: Yes

Reviewer #7: I Don't Know

4. Have the authors made all data underlying the findings in their manuscript fully available?

Reviewer #6: Yes

Reviewer #7: Yes

5. Is the manuscript presented in an intelligible fashion and written in standard English?

Reviewer #6: Yes

Reviewer #7: Yes

Reviewer #6: Thank you for your thoughtful revisions to the manuscript. Although this remains a study with a relatively small sample size, I appreciate that you have carefully addressed the points raised by the previous reviewers. I would kindly ask you to consider the following additional questions and clarifications, which may help to further strengthen the clarity of the manuscript.

Comment 1

According to the Methods, retreatment success was defined as the ability to continue oral mesalamine for more than 6 months. However, in the Results and Fig. 2, some patients classified into the intolerant group appear to have achieved ‘success’ with second- or third-line mesalamine therapy. The subsequent clinical course of these patients (e.g., whether they later discontinued mesalamine due to AEs) is not clearly described. Please clarify how such cases were classified into the final tolerant vs. intolerant groups.

Comment 2

In Results (lines 185) and Figure 2, the phrase “28 patients with AEs” is mentioned. Should this actually refer to the 28 patients who underwent DLST?

Comment 3

In the sentence “All 13 patients with positive F-DLSTs were mesalamine intolerant (p = 0.018, OR 2.444, 95% CI 1.479–4.039) (Table 3),”

it would be clearer to state “mesalamine intolerant group” rather than simply “mesalamine intolerant.”

Comment 4

There is an inconsistency between the study period reported in the Materials and Methods section and that in the Results section. Please revise accordingly.

Reviewer #7: This research paper examines 5-ASA intolerance, a topic that has gained attention in recent years. The content is intriguing. However, as Reviewer 5 noted, unfortunately, the number of patients examined in this study is small. Also, were there no patients among the 22 with mesalamine intolerance who used SASP? Unless the authors evaluated the efficacy of SASP, these patients cannot be considered actual cases of mesalamine intolerance. Please reconsider.

**Do you want your identity to be public for this peer review?** For information about this choice, including consent withdrawal, please see our Privacy Policy

Reviewer #6: No

Reviewer #7: No

---

## [Author Response · Author response to Decision Letter 4]

24 Sep 2025

Manuscript ID: PONE-D-25-17000R3

Title: Significance of the drug-induced lymphocyte stimulation test for various oral mesalamines in ulcerative colitis with mesalamine intolerance

Dear Editors and Reviewers:

We sincerely thank the Academic Editor and Reviewers for their thoughtful and constructive comments, which have been very helpful in improving our manuscript. We have carefully revised the manuscript in accordance with the suggestions, and the specific responses to each comment are detailed below. Page and line numbers refer to the revised version of the manuscript.

Reviewer # 6, Comment 1:

According to the Methods, retreatment success was defined as the ability to continue oral mesalamine for more than 6 months. However, in the Results and Fig. 2, some patients classified into the intolerant group appear to have achieved ‘success’ with second- or third-line mesalamine therapy. The subsequent clinical course of these patients (e.g., whether they later discontinued mesalamine due to AEs) is not clearly described. Please clarify how such cases were classified into the final tolerant vs. intolerant groups.

Response:

We sincerely thank the reviewer for this helpful comment. To address the concern:

1. We revised the Results sentence for clarity. The original wording “Fourteen patients underwent retreatment with oral mesalamines after experiencing AEs in response to the first 5-ASA (Fig 2), including six patients who were tolerant to oral mesalamines and eight patients who were intolerant to oral mesalamines” has been replaced with “Fourteen patients underwent retreatment with oral mesalamines after AEs to the first 5-ASA (Fig 2), of whom six achieved retreatment success and were classified into the mesalamine-tolerant group, whereas eight experienced retreatment failure and were classified into the mesalamine-intolerant group.” (page 13, lines 245–248)

2. We added details of the clinical course for patients with successful retreatment: “Six patients who achieved successful retreatment maintained oral mesalamine therapy at a minimum dose of 2,000 mg/day for at least 6 months, and none discontinued treatment thereafter.” (page 14-15, lines 270–273)

3. To avoid ambiguity, we also added a statement in the Methods explicitly clarifying that patients with successful retreatment were classified into the mesalamine-tolerant group, whereas those with retreatment failure were classified into the mesalamine-intolerant group. (page 6, lines 129–131)

We believe these revisions clearly address the reviewer’s concern and eliminate any potential misunderstanding regarding the classification of patients.

Comment 2:

In Results (line 185) and Figure 2, the phrase “28 patients with AEs” is mentioned. Should this actually refer to the 28 patients who underwent DLST?

Response:

We thank the reviewer for this careful observation. The phrase “28 patients with AEs” was indeed misleading. What we intended to describe was the 28 patients who underwent DLST. We have revised the Results section (page 9, line 186) and the corresponding label in Figure 2 to read “Patients who underwent DLST” to avoid confusion.

Comment 3:

In the sentence “All 13 patients with positive F-DLSTs were mesalamine intolerant (p = 0.018, OR 2.444, 95% CI 1.479–4.039) (Table 3),” it would be clearer to state “mesalamine intolerant group” rather than simply “mesalamine intolerant.”

Response:

We appreciate the reviewer’s helpful suggestion. We agree that “mesalamine-intolerant group” is clearer and avoids potential ambiguity. We have revised the sentence in the Results section (page 12, line 219) to read: “All 13 patients with positive F-DLSTs were in the mesalamine-intolerant group (p = 0.018, OR 2.444, 95% CI 1.479–4.039) (Table 3).”

Comment 4:

There is an inconsistency between the study period reported in the Materials and Methods section and that in the Results section. Please revise accordingly.

Response:

We thank the reviewer for pointing out this inconsistency. We have corrected the discrepancy by unifying the study period throughout the manuscript. The study period is now consistently reported as between 2014 and 2021 in both the Materials and Methods and Results sections. (page 9, line 180)

Reviewer #7:

This research paper examines 5-ASA intolerance, a topic that has gained attention in recent years. The content is intriguing. However, as Reviewer 5 noted, unfortunately, the number of patients examined in this study is small. Also, were there no patients among the 22 with mesalamine intolerance who used SASP? Unless the authors evaluated the efficacy of SASP, these patients cannot be considered actual cases of mesalamine intolerance. Please reconsider.

Response:

We thank the reviewer for these constructive comments.

1. We fully acknowledge the limitation of the relatively small sample size. As noted by Reviewer 5, we have already addressed this point in the Limitations section, emphasizing the need for confirmation in larger, multicenter prospective cohorts.

2. With regard to SASP, we confirm that SASP retreatment was performed and evaluated in this study. In fact, SASP was included among the successful retreatment cases. Specifically, two patients who failed initial 5-ASA therapy subsequently tolerated SASP and maintained treatment without recurrence of adverse events. These patients were therefore classified into the mesalamine-tolerant group. Conversely, three patients experienced retreatment failure with SASP (one in the second-line therapy and two in the third-line therapy). We have revised the Results section (page 14, lines 255–259, 262–265) to make these details explicit, thereby clarifying that the efficacy of SASP was fully assessed in our cohort.

We believe these revisions adequately address the reviewer’s concern.

We carefully reviewed the reviewer comments and confirm that no additional references were specifically recommended in this revision round. Accordingly, no new citations were added.

The revised manuscript with tracked changes, a clean version of the manuscript, and a response letter were uploaded separately.

We confirmed that our protocols do not require registration in protocols.io because they do not involve novel laboratory methods.

We hope that the revisions and clarifications satisfactorily address the concerns of the reviewers. We greatly appreciate the constructive feedback of the reviewers and your consideration of our revised manuscript. We hope that the revised manuscript will be acceptable for publication in PLOS ONE.

Sincerely,

Akira Madarame, MD

---

## [Decision Letter · Decision Letter 4]

6 Oct 2025

Significance of the drug-induced lymphocyte stimulation test for various oral mesalamines in ulcerative colitis with mesalamine intolerance

PONE-D-25-17000R4

Dear Dr. Madarame,

We’re pleased to inform you that your manuscript has been judged scientifically suitable for publication and will be formally accepted for publication once it meets all outstanding technical requirements.

Kind regards,

Dr. Mohammed Misbah Ul Haq, Pharm-D

Academic Editor

PLOS ONE

Additional Editor Comments (optional):

Reviewers' comments:

Reviewer's Responses to Questions

**Comments to the Author**

Reviewer #6: (No Response)

Reviewer #8: All comments have been addressed

2. Is the manuscript technically sound, and do the data support the conclusions?

Reviewer #6: Yes

Reviewer #8: Yes

3. Has the statistical analysis been performed appropriately and rigorously?

Reviewer #6: Yes

Reviewer #8: Yes

4. Have the authors made all data underlying the findings in their manuscript fully available?

Reviewer #6: Yes

Reviewer #8: Yes

5. Is the manuscript presented in an intelligible fashion and written in standard English?

Reviewer #6: Yes

Reviewer #8: Yes

Reviewer #6: Thank you for your clarifications regarding the classification of mesalamine-tolerant and intolerant patients. However, in Fig. 2 the designation of “(–): N=6” is not clearly explained. According to the Methods, patients achieving retreatment success (≥6 months of mesalamine continuation without AEs) should be classified into the tolerant group. Could you please clarify whether the “(–): N=6” patients in Fig. 2 correspond to the mesalamine-tolerant group?

To avoid further confusion, please revise the figure or legend to explicitly indicate which subgroup in Fig. 2 constitutes the final mesalamine-tolerant group (n=6), and how these patients relate to the intermediate “success” branches shown at the second- and third-line retreatments. This will help ensure consistency between the Methods, the Results, and Fig. 2.

Reviewer #8: This paper is potentially interesting and worthy of eventual publication. Your revised paper is well written and illustrated because all comments are taking into account.

**Do you want your identity to be public for this peer review?** For information about this choice, including consent withdrawal, please see our Privacy Policy

Reviewer #6: No

Reviewer #8: No

---

## [Editor Report · Acceptance letter]

PONE-D-25-17000R4

PLOS ONE

Dear Dr. Madarame,

I'm pleased to inform you that your manuscript has been deemed suitable for publication in PLOS ONE. Congratulations! Your manuscript is now being handed over to our production team.

Kind regards,

on behalf of

Dr. Mohammed Misbah Ul Haq

Academic Editor

PLOS ONE